# Short- and Long-Term Humoral and Cellular Immune Responses to SARS-CoV-2 Vaccination in Patients with Multiple Sclerosis Treated with Disease-Modifying Therapies

**DOI:** 10.3390/vaccines11040786

**Published:** 2023-04-03

**Authors:** Susana Sainz de la Maza, Paulette Esperanza Walo-Delgado, Mario Rodríguez-Domínguez, Enric Monreal, Alexander Rodero-Romero, Juan Luis Chico-García, Roberto Pariente, Fernando Rodríguez-Jorge, Rubén Ballester-González, Noelia Villarrubia, Beatriz Romero-Hernández, Jaime Masjuan, Lucienne Costa-Frossard, Luisa María Villar

**Affiliations:** 1Department of Neurology, Hospital Universitario Ramón y Cajal, Instituto Ramón y Cajal de Investigación Sanitaria (IRYCIS), Red Española de Esclerosis Múltiple (REEM), Universidad de Alcalá, 28034 Madrid, Spain; 2Department of Immunology, Hospital Universitario Ramón y Cajal, Universidad de Alcalá, Instituto Ramón y Cajal de Investigación Sanitaria (IRYCIS), Red Española de Esclerosis Múltiple (REEM), 28034 Madrid, Spain; 3Department of Microbiology, Hospital Universitario Ramón y Cajal, Instituto Ramón y Cajal de Investigación Sanitaria (IRYCIS), CIBER en Epidemiología y Salud Pública (CIBERESP), 28034 Madrid, Spain

**Keywords:** SARS-CoV-2 vaccination, immune response, COVID19, multiple sclerosis, disease-modifying therapies

## Abstract

Background: This study aimed to evaluate short- and long-term humoral and T-cell-specific immune responses to SARS-CoV-2 vaccines in patients with multiple sclerosis (MS) treated with different disease-modifying therapies (DMTs). Methods: Single-center observational longitudinal study including 102 patients with MS who consecutively received vaccination against SARS-CoV-2. Serum samples were collected at baseline and after receiving the second dose of the vaccine. Specific Th1 responses following in vitro stimulation with spike and nucleocapsid peptides were analyzed by quantifying levels of IFN-γ. Serum IgG-type antibodies against the spike region of SARS-CoV-2 were studied by chemiluminescent microparticle immunoassay. Results: Patients undergoing fingolimod and anti-CD20 therapies had a markedly lower humoral response than those treated with other DMTs and untreated patients. Robust antigen-specific T-cell responses were detected in all patients except those treated with fingolimod, who had lower IFN-γ levels than those treated with other DMTs (25.8 pg/mL vs. 868.7 pg/mL, *p* = 0.011). At mid-term follow-up, a decrease in vaccine-induced anti-SARS-CoV-2 IgG antibodies was observed in all subgroups of patients receiving DMTs, although most patients receiving induction DMTs or natalizumab and non-treated patients remained protected. Cellular immunity was maintained above protective levels in all DMT subgroups except the fingolimod subgroup. Conclusions: SARS-CoV-2 vaccines induce robust and long-lasting humoral and cell-mediated specific immune responses in most patients with MS.

## 1. Introduction

Over the past decade, the disease course of multiple sclerosis (MS) patients appears to have become milder. This may be partly due to the availability of effective disease-modifying therapies (DMTs) that better reduce inflammatory disease activity and delay disability progression [1]. The various mechanisms of action of the DMTs, including lymphocyte depletion, disruption of lymphocyte replication, or alteration of lymphocyte trafficking, impact the immune system [2,3]. As a result, patients with MS receiving DMTs could not only have an increased risk of infections but also reduced vaccine effectiveness because of a decreased ability to mount an adaptive immune response [4,5].

The COronaVIrus Disease 2019 (COVID-19) pandemic, caused by severe acute respiratory syndrome coronavirus 2 (SARS-CoV-2), is a rapidly evolving situation that continues to cause unprecedented disruption of normal life. As the COVID-19 pandemic spread worldwide, an increasing number of SARS-CoV-2 vaccines were developed. Preliminary data suggest that the humoral response to the SARS-CoV-2 vaccines is impaired in MS patients treated with fingolimod and anti-CD20 therapies [6,7,8]. However, vaccine-induced T-cell responses are believed to play an essential role in protection against subsequent SARS-CoV-2 infections. Accordingly, some studies reported effects of some DMTs on cellular immune responses following vaccination [9,10,11].

We aimed to evaluate the humoral and cellular immune responses to SARS-CoV-2 vaccines in patients with MS receiving different DMTs and to explore their protective role against COVID-19 during patient follow-up.

## 2. Materials and Methods

### 2.1. Study Design

We performed a single-center observational longitudinal study including 102 MS patients who consecutively received vaccination against SARS-CoV-2 at Hospital Universitario Ramón y Cajal in Madrid, Spain. Inclusion criteria were: diagnosis of MS according to McDonald 2017 criteria [12] and completed vaccination cycle of an mRNA or viral-vector SARS-CoV-2 vaccine. Both untreated patients and patients treated with DMTs were included. Approval was obtained from the Ethical Committee of Hospital Universitario Ramón y Cajal. Patients provided written informed consent before inclusion.

### 2.2. Data Collection

At baseline, before vaccination, demographic characteristics, time since first MS symptoms, MS phenotype, disability according to the Expanded Disability Status Scale (EDSS) score, current DMT, time since DMT initiation, last DMT administration (in patients undergoing pulsed therapy), and occurrence of COVID-19 before vaccination were recorded.

Patients were subsequently evaluated every 3 months after vaccination to assess the occurrence of COVID-19.

### 2.3. Sample Collection

Serum samples (5 mL) were collected after vaccination (at the earliest, 28 days after vaccination), aliquoted, and stored at 80 °C until studied. Peripheral blood mononuclear cells (PBMCs) were obtained from heparinized whole blood by Ficoll density gradient centrifugation (Abbott).

### 2.4. Cell Cultures

Fresh PBMCs were cultured in 96-well flat-bottom plates at 10^6^ cells/well resuspended in a volume of 200 µL of complete medium enriched with 10% human serum (Merck). Four culture wells were prepared for each patient: one well with complete medium (negative control), another well with 10 µL (10 µg/mL) of OKT3 as a positive control; the other two wells were stimulated with 4 µL (50 µg/mL) of spike (S) peptides of SARS-CoV-2 (PepTivator^®^ SARS-CoV-2 Prot_S, Miltenyi Biotec) and with 4 µL (50 µg/mL) of nucleocapsid (N) peptides of SARS-CoV-2 (PepTivator^®^ SARS-CoV-2 Prot_N, Miltenyi Biotec). After 30 min, 6 µL (25 µg/mL) of CD28/CD49d co-stimulator was added to each well, and cells were incubated at 37 °C, 5% CO_2_, and 95% humidity for 24 h. Then, the cellular suspension was centrifuged, and supernatants were collected and stored at −80 °C until analysis.

### 2.5. Interferon-Gamma Quantification

Supernatant interferon-gamma (IFN-γ) levels were quantified in a SR-X instrument (Quanterix, Billerica, MA, USA) using the single-molecule array IFN-γ Advantage Kit technique (Quanterix). Since no consensus has been established on the best cut-off value to consider a good cellular response to SARS-CoV-2, we considered a positive result when the concentration of IFN-γ was higher than 80 pg/mL, based on results obtained in our untreated patients.

### 2.6. Serum Anti-Spike Antibodies

Serum IgG-type antibodies against the spike region of SARS-CoV-2 (S1 subunit) were studied by chemiluminescent immunoassay of micro-particles (CMIA) in the automated system ALINITY (Abbott Laboratories, Chicago, IL, USA). Following the instructions for using the WHO International Standard and Reference Panel for anti-SARS-CoV-2 antibody [13,14], anti-spike SARS-CoV-2 antibodies were expressed in binding antibody units per milliliter (BAU/mL) in this study. For this purpose, arbitrary units per mL (AU/mL) were converted to BAU/mL using the conversion factor provided by the manufacturer (1 BAU/mL = 0.142 × AU/mL). Anti-spike IgG antibody titers greater than 50 AU/mL (i.e., 7.1 BAU/mL) were considered positive by the manufacturer, but titers greater than 260 BAU/mL are protective against SARS-CoV-2 disease, based on recent data [15].

### 2.7. Persistence of Immune Response to SARS-CoV-2 Vaccines

To evaluate the persistence of immune responses after two vaccine doses, serum samples obtained from each patient were classified into two groups: samples obtained between 0 and 90 days after the second vaccine dose (early sample) and samples obtained beyond 90 days after the second vaccine dose (late sample). The immune response was considered persistent when determinations were positive before and after 90 days or when a patient without an early sample presented a positive late sample.

### 2.8. Statistical Analysis

Analyses were performed using the GraphPad Prism 6.0 software (GraphPad Prism Inc., La Jolla, CA, USA). Categorical variables were summarized using frequencies (percentages) and were analyzed with the χ^2^ test. Continuous variables were reported as median [interquartile range, IQR] and were analyzed with Wilcoxon rank-sum test. Kruskal–Wallis test was used for between-group comparisons. Associations between demographic, clinical, and laboratory data and humoral and cellular immune responses were assessed using Spearman’s rank correlations. Two-tailed *p*-values < 0.05 were considered significant.

## 3. Results

### 3.1. Patients

One hundred and two patients were prospectively included in the study. Baseline demographic and clinical characteristics are shown in Table 1. Ninety-seven patients were treated with DMTs: 16 with first-line therapies (FLT), 13 with fingolimod, 15 with cladribine, seven with natalizumab, 28 with anti-CD20 therapies, and 18 with alemtuzumab.

Table 2 details the duration of treatment, the lymphocyte counts before the first vaccine dose, and the time elapsed since the last drug dose for anti-CD20 therapies and pulsed immune reconstitution therapies. Lymphocyte counts before the first vaccine dose were lower in fingolimod-treated patients and cladribine-treated patients compared to those treated with other DMTs (*p* = 0.0001 for both comparisons).

Ninety-two patients (90.2%) were vaccinated with both doses of an mRNA vaccine (mRNA-1273 or BNT162b2), seven patients (6.9%) were vaccinated with both doses of ChAdOx1nCoV-19, and three patients (2.9%) were vaccinated with a dose of Ad26.COV2-S and then a dose of an mRNA vaccine. There were no statistically significant differences between groups in demographic, clinical, and treatment characteristics. Fifty-two patients (51%) received a third dose of an mRNA vaccine. However, in all cases, the immune response was evaluated before the third dose.

### 3.2. Humoral Immune Response

Fourteen (22.6%) of the 62 patients with available pre-vaccination serological analysis were positive for IgG anti-SARS-CoV-2 antibodies as a result of previous infection, but only two had protective titers. After two doses of a SARS-CoV-2 vaccine, 84 (82.4%) patients had positive anti-SARS-CoV-2 IgG titers, and 65 (63.7%) of them reached protective titer levels. 

Post-vaccination anti-SARS-CoV-2 IgG titers by DMTs are represented in Figure 1A. Patients undergoing fingolimod and anti-CD20 therapies had a markedly lower humoral response than those treated with other DMTs. Eight of 13 (61.5%) patients receiving fingolimod had positive anti-SARS-CoV-2 IgG titers, and three of 13 (23.1%) achieved protective levels. The median anti-SARS-CoV-2 IgG titer in patients with fingolimod was 21.5 [3.4–160.6] BAU/mL in comparison with 717.8 [87.6–2287.8] BAU/mL in patients with other DMTs (*p* = 0.0001). Likewise, seven of 16 (44%) patients under ocrelizumab had positive anti-SARS-CoV-2 IgG titers, and three of 16 (19%) had protective levels. The median titer was 6.0 [1.2–231.8] BAU/mL, clearly lower than those in patients with other DMTs (*p* = 0.0001). Similarly, eight of 12 (66.7%) patients receiving rituximab had positive anti-SARS-CoV-2 IgG titers, and five of 12 reached protective levels. The median titer of 106.3 [4.5–384.4] BAU/mL was again lower than those in patients with other DMTs (*p* = 0.0001).

### 3.3. Cellular Immune Response

Thirteen (21%) of the 62 patients with available pre-vaccination IFN-γ quantification had levels ≥80 pg/mL due to a previous infection. After two doses of a SARS-CoV-2 vaccine, 81 of the 92 patients with an IFN-γ determination had levels ≥80 pg/mL, with a median value of 876.9 [270.7–2764] pg/mL.

Post-vaccination IFN-γ levels in patients classified according to their DMTs are represented in Figure 1B. Patients with fingolimod had lower IFN-γ levels than those receiving other DMTs (25.8 pg/mL vs. 868.7 pg/mL, *p* = 0.011). There were no other significant differences between DMTs.

The concordance between cellular and humoral responses is represented in Figure 2. Thirteen patients receiving ocrelizumab, seven receiving rituximab, and seven receiving fingolimod who had anti-SARS-CoV-2 IgG titers < 260 BAU/mL had a positive cellular response with IFN-γ levels ≥ 80 pg/mL. We found no correlation between anti-SARS-CoV-2 IgG titers and IFN-γ levels (*rho* = 0.18, *p* = 0.08).

### 3.4. Persistence of Humoral and Cellular Immune Responses

The median time between the second dose of the vaccine and sample extraction was 42 [28.5–51.5] days for the early samples and 140 [124–177] for the late samples. After two doses of a SARS-CoV-2 vaccine in the early samples, 53/80 (66.3%) patients had anti-SARS-CoV-2 IgG titers ≥ 260 BAU/mL, and 66/75 (88%) had IFN-γ levels ≥ 80 pg/mL. Twenty-six of the 41 patients (63.4%) with a late sample showed persistent protective levels of IgG anti-SARS-CoV-2 antibodies, and 18/22 (81.8%) IFN-γ levels higher than 80 pg/mL. When quantitative values were explored, we observed higher antibody titers in early (*n* = 89) compared to late (*n* = 40) samples (855.6 [218.6–2549] BAU/mL vs. 269.7 [132.4–977.3] BAU/mL, respectively, *p* = 0.03). Likewise, values of IFN-γ decreased from 855.6 [218.6–2549] pg/mL in the early samples to 269.7 [132.4–977.3] pg/mL in the late samples (*p* = 0.005).

We next studied changes in patients classified according to DMTs. Results are shown in Figure 3. Antibody levels were moderately lower in the late samples of patients treated with natalizumab (*p* = 0.03), alemtuzumab (*p* = 0.04), or FLT (*p* = 0.005). No significant decreases were observed in untreated patients or those treated with cladribine. Finally, levels were low in early or late samples of most patients treated with anti-CD20 antibodies or fingolimod.

Moreover, in most patients receiving induction therapies or natalizumab and in non-treated patients, anti-SARS-CoV-2 IgG titers remained above protective levels (Figure 3A). In the case of the cellular immune response, all patients except those treated with fingolimod showed persistent protective IFN-γ levels in the late sample (Figure 3B). In some cases, patients treated with fingolimod had an early protective cellular response, but median IFN-γ levels decreased from 201.42 to 9.47 pg/mL (*p* = 0.04) in the late samples.

### 3.5. Variables Affecting Humoral and Cellular Immune Response to SARS-CoV-2 Vaccines

Demographic, clinical, and laboratory data that could affect vaccines’ immune response were studied. Age weakly correlated with post-vaccination anti-SARS-CoV-2 IgG titers (*rho* = −0.32, *p* = 0.001) but not with IFN-γ levels (*rho* = 0.13, *p* = 0.22). Disability measured with the EDSS score did not correlate with either humoral or cellular immune responses. Absolute lymphocyte counts classified into three grades (>800 cells/μL, 500–799 cells/μL, and <500 cells/μL) in relation to post-vaccination anti-SARS-CoV-2 IgG titers and IFN-γ levels by DMT are presented in Figure 4. There was no correlation between humoral or cellular immune response and the lymphocyte counts in patients treated with fingolimod, cladribine, alemtuzumab, or anti-CD20 therapies.

For patients under treatment with alemtuzumab, cladribine, or anti-CD20 therapies, the effect of the time elapsed between the last dose and the vaccination on the immune response to SARS-CoV-2 was analyzed. We found no correlation between humoral or cellular immune responses and the time elapsed since the last drug dose. No significant differences were found in anti-SARS-CoV-2 IgG titers or IFN-γ levels between relapsing and progressive MS patients. Along the same lines, we found no statistical differences in the humoral or cellular responses of the 20 patients treated with anti-CD20 therapies who had a progressive course and the 11 with a relapsing course.

Since prior SARS-CoV-2 infection influences immune responses to SARS-CoV-2 vaccines, we separately analyzed patients recovered from COVID-19 and observed that they had significantly higher post-vaccination anti-SARS-CoV-2 IgG titers than patients without prior COVID-19 (2329 vs. 439.4 BAU/mL, *p* = 0.005). When we analyzed pre-vaccination N-reactive T-cell responses, patients recovered from COVID-19 had significantly higher IFN-γ levels than patients without prior COVID-19. After receiving a full course of a SARS-CoV-2 vaccine, IFN-γ levels progressively decreased in both groups of patients, although they remained statistically higher in patients with previous COVID-19 (Figure 5A). Meanwhile, IFN-γ levels measured after in vitro stimulation with S-peptide showed a progressive increase after vaccination in both previously infected and uninfected patients (Figure 5B), with no statistical difference between them.

Finally, we analyzed the influence of COVID-19 vaccine type on vaccination-induced immune responses and found no differences in anti-SARS-CoV-2 IgG titers or IFN-γ levels between vaccine types.

### 3.6. Association between Anti-SARS-CoV-2 IgG titers, IFN-γ T-Cell-Specific Response, and Subsequent COVID-19

During a median follow-up of 303 [IQR 253–315] days after receiving the first dose of a SARS-CoV-2 vaccine, 14 patients suffered post-vaccination COVID-19: two patients with no DMT, three of 16 patients with FLT, two of 16 with ocrelizumab, two of 14 with rituximab, two of 15 with cladribine, one of 13 with fingolimod, two of 18 with alemtuzumab, and none of seven patients with natalizumab. There was no significant difference in COVID-19 rates between these groups. The only patient with COVID-19 requiring hospitalization for oxygen and dexamethasone after vaccination was on fingolimod and had very low post-vaccination anti-SARS-CoV-2 IgG titers and only moderate IFN-γ production. The remaining cases of COVID-19 were mild, regardless of the DMT they received.

We found no association between anti-spike SARS-CoV-2 titers or IFN-γ levels and subsequent COVID-19, but all cases occurred more than 3 months after the second vaccine dose, when humoral and cellular immune responses tend to be lower.

## 4. Discussion

We analyzed the humoral and cellular immune responses to SARS-CoV-2 vaccination and the long-term persistence of the immune responses in MS patients who received different DMTs from the available therapeutic arsenal. In addition, we studied the protective role of these vaccine-induced immune responses against subsequent COVID-19 during a one-year observation period.

Although neutralizing antibodies are associated with protective immunity against subsequent COVID-19, vaccine-induced T-cell responses are believed to play an essential role in the prevention of following SARS-CoV-2 infections [16]. Since the availability of COVID-19 vaccination, there have been studies that reported increasing evidence of specific anti-SARS-CoV-2 antibody responses in MS patients receiving different DMTs [6,7,8]. However, they have some limitations, as they mainly studied a few DMTs. In addition, there are differences or contradictory results related to the sensitivity of the assays used to detect the specific anti-SARS-CoV-2 antibodies and the units used to express antibody titers. Furthermore, only a few studies reported the effects of some DMTs on cellular immune responses upon vaccination [10,17,18,19]. Finally, the persistence of the immune response and its protective role in the prevention of COVID-19 remains to be clearly established. 

This study demonstrated normal humoral and T-cell responses to COVID-19 vaccination in patients treated with FLT, natalizumab, cladribine, and alemtuzumab. In fact, these groups of patients showed anti-SARS-CoV-2 IgG titers and IFN-γ levels similar to those reported in the literature for healthy individuals [10,20]. In contrast, as previously described, we found lower anti-spike IgG SARS-CoV-2 titers in patients who received fingolimod, rituximab, and ocrelizumab, compared to patients treated with the other DMTs and non-treated patients. However, robust antigen-specific T-cell responses were detected in all patients except those treated with fingolimod. This effect was independent of the time from the last treatment administration in the case of anti-CD20 therapies, cladribine and alemtuzumab, and was also independent of the absolute lymphocyte count at the moment of vaccination.

Previous SARS-CoV-2 infection has been reported to positively influence vaccine-induced antibody responses in the general population [21,22,23]. Our study demonstrated that this humoral immune response exhibits similar dynamics in MS patients, showing that COVID-19-recovered vaccinees had significantly higher antibody titers than SARS-CoV-2-naïve vaccinees. Response against N-peptide was higher in previously infected than in uninfected vaccinated patients, but pre-vaccination IFN-γ levels decreased in post-vaccination samples, implying that infection-acquired immune responses are lost over time. However, consistent with previous reports in healthcare workers [23], no differences in vaccine-induced S-reactive T-cell responses were observed between previously infected and uninfected vaccinated patients, confirming that vaccines are capable of mounting strong and long-lasting T-cell immune responses in both groups. Indeed, since S protein is the target of the mRNA and adenovirus-based vaccines used in our study, analysis of S-reactive T-cell responses allows monitoring of vaccine-induced immunity but does not distinguish between immune responses triggered by vaccination and those induced by infection. In contrast, N-reactive T-cell responses can be observed in COVID-19-recovered vaccinees but not in SARS-CoV-2-naïve vaccinees, thus allowing differentiation between vaccinated individuals with or without prior COVID-19.

It is widely believed that antibodies can be a useful surrogate marker of CD4+ T-cell responses after most vaccinations. It is difficult to measure virus-specific T cells on a massive scale, so correlations between antibodies and T cells are of great interest [16]. However, in the specific case of COVID-19 patients, a recent study found that anti-SARS-CoV-2 IgG titers were not a surrogate indicator of the magnitude of memory T cells [24], suggesting that simple antibody diagnostic tests are not a robust indicator of protective immunity in people previously infected with SARS-CoV-2. Our study also found no correlation between anti-SARS-CoV-2 IgG titers and IFN-γ levels. This is important in patients treated with antiCD20 antibodies, which had been considered to have a defective response to the virus based on antibody data [6,17] but have a good cellular response as validated here.

We also analyzed the persistence of the immune response to SARS-CoV-2 vaccines in patients with MS. This is important since new waves of the infection arise periodically, and although this could be attributed to the new variants, a decrease in the effector immune response could also contribute to this periodic rebound in new cases. Our study revealed a decrease in vaccine-induced anti-SARS-CoV-2 IgG antibodies at mid-term follow-up in all subgroups of DMTs, but most patients who received induction DMTs or natalizumab and non-treated patients remained protected. A decreasing immune response several months after the second dose of BNT162b2 has been described in a longitudinal observational study in healthy employees at a German hospital [20] and a retrospective study of COVID-19-confirmed convalescent patients with MS [25]. 

Interestingly, we showed that cell-mediated immunity is maintained above protective levels in all DMT subgroups except fingolimod. This could account for the low proportion of MS patients (14 out of 102) suffering a subsequent COVID-19 and the mild disease course observed in most cases. In our study, the only patient who required hospitalization was on fingolimod treatment and had not developed humoral or cellular protective responses after SARS-CoV-2 vaccination. Moreover, 3 months after the second vaccination (when antibody and cellular immune responses were high), no patients suffered COVID-19.

No association between SARS-CoV-2 titers or IFN-γ levels and subsequent COVID-19 was demonstrated in our study. This may be because most of our patients with post-vaccination COVID-19 were infected with the Omicron variant, given their symptoms and the timing of their presentation. Results from a recently published study [26] show that vaccine efficacy against symptomatic disease is lower for Omicron, so more vaccinated individuals are likely to develop COVID-19 due to Omicron [27]. However, this could also be due to the time at which we monitored the immune response, in the period of 1 and 3 months after the second vaccine dose in most patients. This could indicate that humoral and cellular immune response assessed in the first trimester after vaccination do not give an idea of later protection against COVID-19 infection. This should be considered for patients at higher risk (SPMS or older age). However, the maintenance of protective cellular response could account for the benign course of new COVID-19 infections, except in cases of patients treated with fingolimod. 

Our study has some limitations. First, the small size of the different cohorts of DMTs restricts the comparison of vaccine-induced immune responses between groups. Second, the timing of sample collection after vaccination was not homogeneous among patients. Despite these limitations, we believe that our findings are relevant to expanding knowledge about immune responses to COVID-19 vaccination and subsequent protection against SARS-CoV-2 infection.

## Figures and Tables

**Figure 1 vaccines-11-00786-f001:**
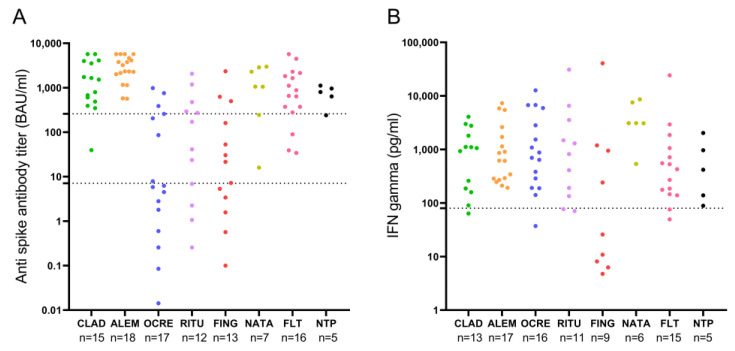
Post-vaccination anti-SARS-CoV-2 IgG titers (**A**) and IFN-γ levels (**B**) by disease-modifying therapy. (**A**) The lower dotted line marks anti-spike IgG antibody titers considered positive (7.1 BAU/mL), and the upper dotted line marks anti-spike IgG antibody titers considered protective (260 BAU/mL). (**B**) Dotted line marks IFN-γ levels of 80 pg/mL that were considered positive. ALEM: alemtuzumab; CLAD: cladribine; FING: fingolimod; FLT: first-line therapies; NATA: natalizumab; NTP: non-treated patients; OCRE: ocrelizumab; RITU: rituximab.

**Figure 2 vaccines-11-00786-f002:**
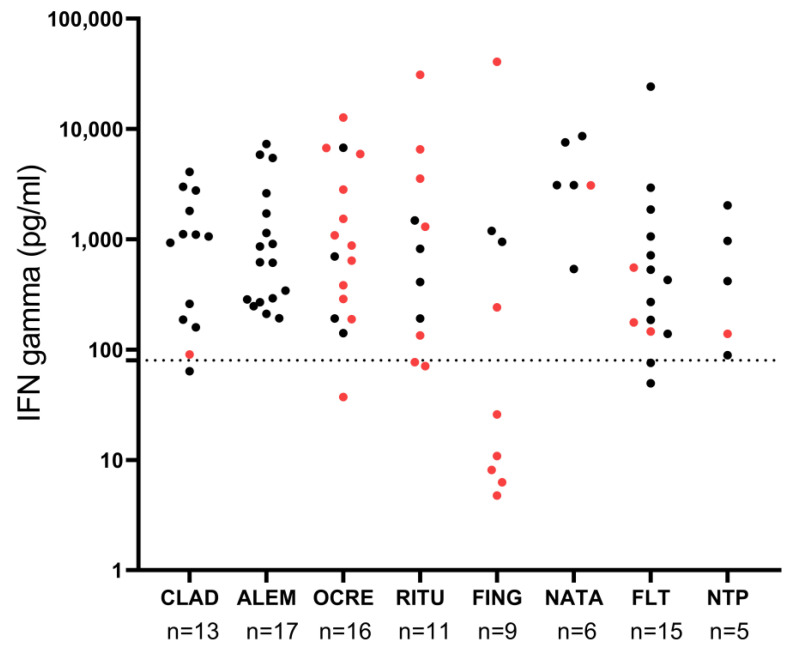
Concordance between cellular and humoral responses. Black dots indicate patients who developed a protective humoral response. Red dots indicate patients who did not develop a protective humoral response. Dotted line marks IFN-γ levels of 80 pg/mL that were considered positive. ALEM: alemtuzumab; CLAD: cladribine; FING: fingolimod; FLT: first-line therapies; NATA: natalizumab; NTP: non-treated patients; OCRE: ocrelizumab; RITU: rituximab.

**Figure 3 vaccines-11-00786-f003:**
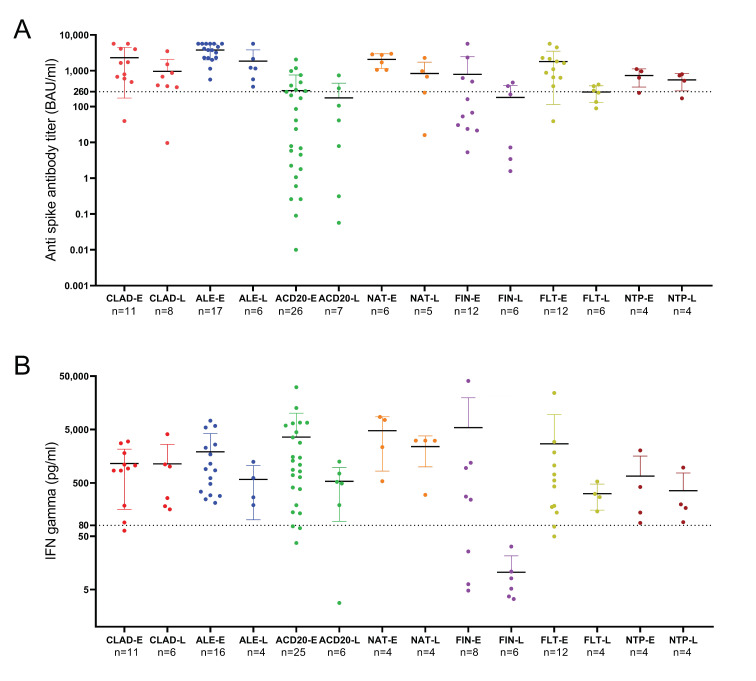
Anti-SARS-CoV-2 IgG titers (**A**) and IFN-γ levels (**B**) in the early and late samples of patients classified according to their DMTs. (**A**) Dotted line marks anti-spike IgG antibody titers considered protective (260 BAU/mL). (**B**) Dotted line marks IFN-γ levels of 80 pg/mL that were considered positive. ACD20: anti-CD20 therapies; ALE: alemtuzumab CLAD: cladribine; E: early sample; FIN: fingolimod; FLT: first-line therapies; L: late sample; NAT: natalizumab; NTP: non-treated patients.

**Figure 4 vaccines-11-00786-f004:**
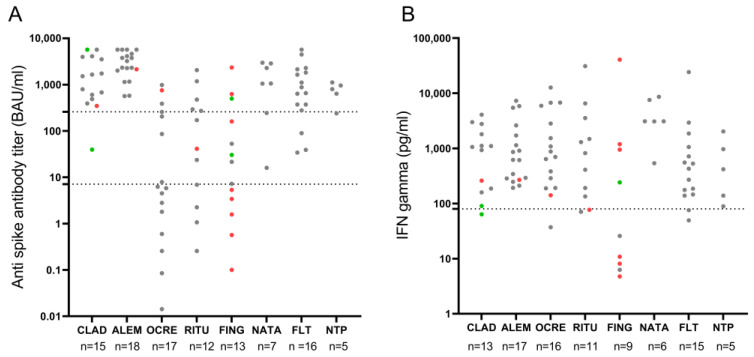
Post-vaccination anti-SARS-CoV-2 IgG titers (**A**) and IFN-γ levels (**B**) in relation to absolute lymphocyte counts: >800 cells/μL (gray dot), 500–799 cells/μL (red dot) and <500 cells/μL (green dot). (**A**) The lower dotted line marks anti-spike IgG antibody titers considered positive (7.1 BAU/mL), and the upper dotted line marks anti-spike IgG antibody titers considered protective (260 BAU/mL). (**B**) Dotted line marks IFN-γ levels of 80 pg/mL that were considered positive. ALEM: alemtuzumab; CLAD: cladribine; FING: fingolimod; FLT: first-line therapies; NATA: natalizumab; NTP: non-treated patients; OCRE: ocrelizumab; RITU: rituximab.

**Figure 5 vaccines-11-00786-f005:**
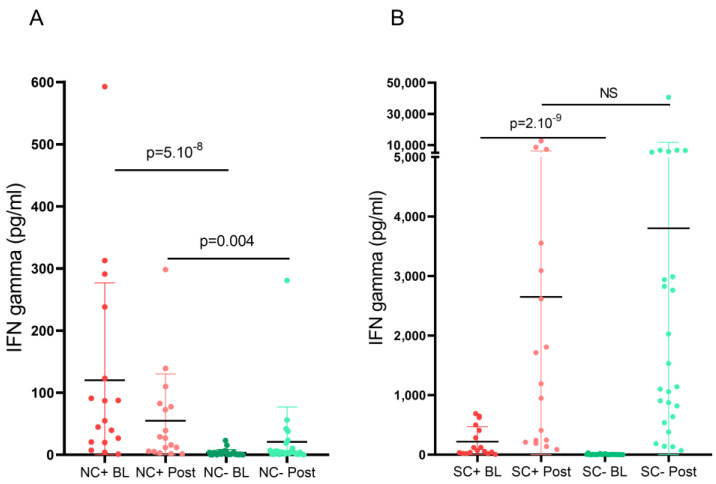
IFN-γ levels measured after in vitro stimulation of nucleocapsid (**A**) and spike peptides (**B**), in baseline and post-vaccination samples, in patients with or without prior COVID19. BL: baseline sample; C+: prior COVID-19; C−: without prior COVID-19; N: nucleocapsid; NS: non-significant; POST: post-vaccination sample; S: spike.

**Table 1 vaccines-11-00786-t001:** Demographic and clinical characteristics.

Characteristics	Total Population (*n* = 102)
Age, median [range] (years)	40.2 [21.1–72.7]
Females, *n* (%)	64 (62.7)
Time since first MS symptoms, median [range]	8.6 [0.2–35.7]
MS phenotype, *n* (%)	
Relapsing–remitting	82 (80.4)
Secondary progressive	14 (13.7)
Primary progressive	6 (5.9)
EDSS score, median [range]	2.5 [1–8.0]
DMT, *n* (%)	
None	5 (4.9)
First-line therapies	16 (15.7)
Fingolimod	13 (12.7)
Cladribine	15 (14.7)
Natalizumab	7 (6.9)
Ocrelizumab	16 (15.7)
Rituximab	12 (11.8)
Alemtuzumab	18 (17.6)
SARS-CoV-2 vaccine, *n* (%)	
BNT162b2 (Pfizer, New York, NY, USA/BioNTech, Mainz, Germany)	72 (70.6)
mRNA-1273 (Moderna, Cambridge, MA, USA)	20 (19.6)
AZD1222 (AstraZeneca, Cambridge, UK)	7 (6.9)
JNJ78436735 (Johnson & Johnson, New Brunswick, NJ, USA)	3 (2.9)

DMT: disease modifying therapy; EDSS: expanded disability status scale; MS: multiple sclerosis.

**Table 2 vaccines-11-00786-t002:** Treatment characteristics at the first vaccine dose.

DMT	Treatment Duration, Median [Range] (Years)	Lymphocyte Count, Median [Range] × 10^3^ Cells/μL	Time Since Last Infusion, Median [Range] (Months)
First-line therapies	4.7 [1.2–14.8]	2.06 [0.95–2.97]	
Fingolimod	6.4 [1.7–9.5]	0.58 [0.37–1.34]	
Cladribine	0.52 [0.2–2.0]	0.99 [0.39–1.60]	4.9 [1.4–20.0]
Natalizumab	2.2 [0.2–8.6]	4.18 [2.24–5.66]	
Ocrelizumab	1.6 [0.4–4.2]	1.75 [0.55–2.44]	3.8 [2.2–8.0]
Rituximab	2.1 [0.3–4.2]	1.89 [0.59–2.69]	6.5 [3.6–23.2]
Alemtuzumab	3.6 [0.2–5.7]	1.55 [0.66–2.77]	28.7 [2.5–55.1]

DMT: disease modifying therapy.

## Data Availability

The data presented in this study are available on request from the corresponding author.

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
