# Peer review of "Short- and Long-Term Humoral and Cellular Immune Responses to SARS-CoV-2 Vaccination in Patients with Multiple Sclerosis Treated with Disease-Modifying Therapies"

_vaccines, 2023, doi:10.3390/vaccines11040786_

Round 1
Reviewer 1 Report
Comments to “Short and long-term humoral and cellular immune responses to SARS-CoV-2 vaccination in patients with multiple sclerosis treated with disease-modifying therapies”In this manuscript, Susana Sainz de la Maza and colleagues report the one-year long comprehensive immune responses induced by SARS-CoV-2 vaccination in multiple sclerosis (MS) patients. MS is a potentially disabling disease of the brain and spinal cord, which usually thought to be caused by the immune system attacks. To fight against MS, several DMTs have approved for clinical treatments. However, some of these DMTs mainly targeting on relative immunology pathways, which may carry significant health risks. In this manuscript, authors analyzed the humoral and cellular immune responses to SARS-CoV-2 vaccination and the long-term persistence of the immune responses in MS patients who received different DMTs. The results shows that SARS-CoV-2 vaccines induce robust and long-lasting humoral and cell-mediated specific immune responses in most patients with MS, while fingolimod (S1P inhibitor) treatment have an effect on vaccination induced protective immune responses. This work provide insight into the adjustment of MS treatment strategies under COVID-19 pandemic. There are some comments below for author’s consideration:
1) As authors mentioned in line 262, 14 patients suffered a post-vaccination COVID-19, is there any available clinical surveillance data to further explain the positive/negative effect of DMTs in COVID-19?
2) To measure the cellular immune response, PBMCs were stimulated by Spike/N protein in Fig1-4, and stimulated by Spike/N peptides in Fig5. What’s the difference in these two methods? Peptides are more commonly used in IFN-γ stimulation.
3) Please check and label the “significant difference marker” in all figures and describe the details in figure legends.
4) In Fig 4, change the gray dot/ red square/ red triangle into the dot marker with three different colors, which is more intuitive.
5) Line 80, “106 cells”, 106 cells.
Reviewer 2 Report
This paper evaluated the humoral and cellular immune responses to SARS-CoV-2 vaccines in multiple sclerosis patients. Cellular immunity was maintained above protective levels in all DMTs subgroups except the fingolimod subgroup. Patients undergoing fingolimod and anti-CD20 therapies had a markedly lower humoral response than those treated with other DMTs and untreated patients. These findings expanding our knowledge about immune responses to COVID-19 vaccination and subsequent protection against SARS-CoV-2 infection, especially in MS patients. This manuscript might be accepted after minor revision.
1. It was good to divide the 102 patients into 8 groups including ALEM, CLAD, FING, FLT, NATA, NTP, OCRE and RITU groups. Besides the non-treated patient group, it is better to add another negative control group—the normal person group. After compared the anti-SARS-CoV-2 IgG titers and IFN-γ levels between normal person group and the MS patient groups, the conclusion “SARS-CoV-2 vaccines induce robust and long-lasting humoral and cell-mediated specific immune responses in most patients with MS” would be made. The author could cite any published paper including the data of anti-SARS-CoV-2 IgG titers and IFN-γ levels for the normal immunized persons, or perform an immune test for the normal persons.
2. It is interesting that the response against N-peptide was higher in previously infected than in uninfected vaccinated patients, while no differences in vaccine-induced S-reactive T-cell responses were observed between previously infected and uninfected vaccinated patients. Did this related to the immunized vaccines including the mRNA vaccine (BNT162b2 and mRNA-1273) and the adenovirus-based vaccine (AZD1222 and JNJ78436735) all based on the S protein? The author can add some discussions in the manuscript.
Reviewer 3 Report
The paper analyzes the effect of COVID vaccines in a cohort of treated and untreated pwMS, evaluating humoral and cell-based immunological response. Patients treated with anti-CD20 and fingolimod have a lower antibody response, while fingolimod seems to determine a lower T-cell response. These results are aligned with avalaible literature to date. Nevertheless, study is well designed and clearly described. I have only a minor request: did you find some different effects in terms of humoral/cellular response stratifying patients according to type of vaccines used? For example, Sormani et al. (eBiomedicine, 2021) found a stronger humoral response after Moderna vaccine.
